# Akt Signaling Pathway Is Activated in the Minor Salivary Glands of Patients with Primary Sjögren’s Syndrome

**DOI:** 10.3390/ijms222413441

**Published:** 2021-12-14

**Authors:** Ioanna E. Stergiou, Loukas Chatzis, Asimina Papanikolaou, Stavroula Giannouli, Athanasios G. Tzioufas, Michael Voulgarelis, Efstathia K. Kapsogeorgou

**Affiliations:** 1Department of Pathophysiology, School of Medicine, National and Kapodistrian University of Athens, 11527 Athens, Greece; stergiouioa@med.uoa.gr (I.E.S.); lukechatzis@gmail.com (L.C.); agtzi@med.uoa.gr (A.G.T.); mvoulgar@med.uoa.gr (M.V.); 2Department of Hematopathology, Evangelismos Hospital, 11527 Athens, Greece; aspapanikolaou@yahoo.com; 3Hematology Unit, Second Department of Internal Medicine, School of Medicine, National and Kapodistrian University of Athens, 11527 Athens, Greece; sgiannoul@med.uoa.gr

**Keywords:** Sjögren’s syndrome, AKT signaling pathway, non-Hodgkin lymphoma, minor salivary glands, salivary gland epithelial cells, infiltrating mononuclear cells

## Abstract

Primary Sjögren’s syndrome (pSS) is an autoimmune exocrinopathy of mainly the salivary and lacrimal glands associated with high prevalence of lymphoma. Akt is a phosphoinositide-dependent serine/threonine kinase, controlling numerous pathological processes, including oncogenesis and autoimmunity. Herein, we sought to examine its implication in pSS pathogenesis and related lymphomagenesis. The expression of the entire and activated forms of Akt (partially and fully activated: phosphorylated at threonine-308 (T308) and serine-473 (S473), respectively), and two of its substrates, the proline-rich Akt-substrate of 40 kDa (PRAS40) and FoxO1 transcription factor has been immunohistochemically examined in minor salivary glands (MSG) of pSS patients (*n* = 29; including 9 with pSS-associated lymphoma) and sicca-complaining controls (sicca-controls; *n* = 10). The entire and phosphorylated Akt, PRAS40, and FoxO1 molecules were strongly, uniformly expressed in the MSG epithelia and infiltrating mononuclear cells of pSS patients, but not sicca-controls. Morphometric analysis revealed that the staining intensity of the fully activated phospho-Akt-S473 in pSS patients (with or without lymphoma) was significantly higher than sicca-controls. Akt pathway activation was independent from the extent or proximity of infiltrates, as well as other disease features, including lymphoma. Our findings support that the Akt pathway is specifically activated in MSGs of pSS patients, revealing novel therapeutic targets.

## 1. Introduction

Primary Sjögren’s syndrome (pSS) is a chronic systemic autoimmune disease with multiple clinical phenotypes extending from mild, limited to exocrine glands, to severe multi-systemic and occasionally life-threatening disease. The exocrinopathy mainly involves the salivary and lachrymal glands and is associated with dry mouth and eyes (sicca symptoms). B-cell non-Hodgkin lymphoma (NHL) develops in 6–10% of patients. The majority of NHLs are derived from mucosa-associated lymphoid tissue (MALT) and located mainly in the affected salivary glands [1,2]. The inflammatory lesions in the minor salivary glands (MSG) follow the heterogeneity of the clinical picture, varying from mild periductal infiltrates with T-cell predominance to severe lesions associated with loss of tissue architecture and B-cell-driven responses [3]. The exact pathogenetic mechanisms involved in the expression of different pSS phenotypes and the progression to lymphomagenesis are not yet clearly defined. Severe MSG inflammatory responses have been associated to NHL development, suggesting a link between local immune responses and the clinical picture of the disease [3,4,5,6,7,8]. It appears that salivary gland epithelial cells (SGEC) studied in MSG biopsies and long-term cultures are not only the target of autoimmune responses, but also the initiators and key regulators of the autoimmune lesion, holding a central role in disease pathogenesis (reviewed in [9]). SGECs in pSS appear to be intrinsically activated, possess features of antigen-presenting cells, and are capable of fruitfully interacting with inflammatory cells [9]. However, the pathways underlying epithelial cell activation in pSS, as well as the mechanisms regulating the interaction between epithelial and immune cells and B-cell transition to lymphoma, are still elusive.

Akt is a phosphoinositide-dependent serine/threonine kinase that regulates diverse physiologic processes in response to extracellular signals provided by growth factors, cytokines, and other stimuli [10,11]. The binding of such stimuli to receptor tyrosine kinases (RTKs) and subsequent activation of phosphatidylinositol-3 kinase (PI3K) results in a signaling cascade leading at first to the partial activation of Akt (phosphorylation of threonine-308 (T308)) and then to its full activation by the phosphorylation of serine-473 (S473) by the mammalian target of Rapamycin (mTOR) complex-2 (mTORC2). Fully activated Akt phosphorylates a variety of substrates implicated in transcription, translation and protein synthesis, cell proliferation, survival and apoptosis, cytoskeletal remodeling and cell migration, autophagy, metabolism, and immune responses. Akt substrates include the proline-rich Akt substrate of 40 kDa (PRAS40), which is a component of the mTOR complex-1 (mTORC1), FoxO transcription factors, p21 cyclin-dependent kinase inhibitor, caspase 9, nuclear transcription factor-κB (NF-κB), etc. [12,13]. 

The activated PI3K/Akt/mTOR pathway has been implicated in various autoimmune diseases [14,15] and it is a common finding in cancer, including hematologic malignancies [16,17]. Its significance in oncogenesis is highlighted by the plethora of inhibitors (dual PI3K–mTOR, PI3K, Akt and mTOR inhibitors) that are in clinical use or late-stage development for cancer treatment, with promising results in hematologic malignancies [17,18]. Thus, Akt could be implicated in pSS pathogenesis and related lymphomagenesis. Little is known about the function of the PI3K/Akt/mTOR signaling pathway in pSS. Epidermal growth factor (EGF) has been shown to activate the PI3K-Akt pathway in cultured SGECs [19], whereas PI3K-Akt activation has been involved in the TLR3-induced apoptosis of cultured SGECs [20]. In addition, upregulated expression of the phosphorylated ribosomal S6 protein (pS6), an end-result of PI3Kdelta signaling, has been detected in the infiltrating T and B lymphocytes at the MSG lesions of pSS patients [21,22]; nevertheless, this was without specifying if this was mediated by the Akt/mTOR or Ras/ERK pathways. 

Prompted by these findings, we sought to investigate the expression and activation of the Akt/mTOR pathway in pSS and possible associations with distinct clinical phenotypes and NHL. Therefore, the expression of the entire and phosphorylated protein forms of Akt kinase and two of its substrates, namely the FoxO1 transcription factor and PRAS40, was immunohistochemically investigated in MSG tissues obtained from pSS patients with variable histological and clinical pictures. 

## 2. Results

### 2.1. The Molecules Related to Active Akt Pathway Are Expressed in pSS Patients, but Not in Sicca-Controls

The entire and phosphorylated forms of Akt and its substrates were found to be strongly expressed in the epithelial and mononuclear cells of the positive control tissue, the stomach biopsies from patients with gastric MALT lymphoma associated with Helicobacter pylori infection (Figure 1). At the MSGs of all pSS patients, Akt and its phosphorylated forms (T308-phosphorylated Akt and S473-phosphorylated Akt, corresponding to partially and fully activated Akt, respectively) were detected in the cytoplasm of ductal and acinar epithelial cells, as well as in all infiltrating mononuclear cells (MNC) (Figure 1 and Appendix A). In 2 out of the 29 pSS patients, 1 with mild lesions and at low risk to develop lymphoma and 1 with NHL, the staining of all Akt forms was low (data not shown and Figure 2A,B). The staining of Akt downstream substrates, namely FoxO1 transcription factor and PRAS40, as well as of their phosphorylated forms, followed a similar pattern with the Akt protein forms. Strong cytoplasmic staining was detected in the epithelial and all infiltrating MNCs at the MSGs of all pSS patients (except the two with low Akt expression) and it was not found to associate with the extent of infiltrates, their organization into ectopic GCs, the predominating immune cell type response, or lymphoma (Figure 1). Furthermore, the staining pattern of all molecules in epithelia was similar throughout the section and not associated with any neighboring inflammatory infiltrates, their size and organization, as well as fibrosis or fat infiltration. On the contrary, Akt and its downstream molecules were not detected in eight out of ten sicca-controls (Figure 1). In the two positive sicca-controls, the Akt (pan and phospho-forms) staining was low and accompanied by low FoxO1 and PRAS40 expression (data not shown and Figure 2A).

### 2.2. The Activation of the Akt Pathway in pSS Does Not Associate with Histologic, Clinical, or Other Disease Features, including NHL Development

To examine possible associations between the extent of Akt pathway activation and patients’ clinical phenotypes, the staining intensity of the fully activated S473-phosphorylated Akt was quantified according to a previously published protocol [21]. Kruskal–Wallis analysis revealed that the staining intensity of S473-phosphorylated Akt normalized by the nuclei intensity value was significantly increased in pSS patients without or with NHL compared to sicca-controls (mean intensity score 8.60, 7.45, and 0.53 in pSS patients without NHL, with NHL, and sicca-controls, respectively; statistical significance: *p* < 0.0001 for pSS patients without NHL vs. sicca-controls and *p* = 0.0026 for pSS patients with NHL vs. sicca-controls) (Figure 2A). However, it was not found to differ between pSS patients with or without lymphoma, or among pSS patients at low or high risk of developing NHL, as well as among pSS patients with a distinct degree of MSG infiltration, namely, mild, intermediate, or severe lesions, whereas it also did not correlate with the biopsy focus score (Figure 2). Furthermore, the normalized S473-phosphorylated Akt staining intensity was not found to be differentially expressed and/or associated with any of the other demographic, histologic, laboratory, and clinical disease parameters examined, including age, disease duration, formation of ectopic GCs at the MSG inflammatory lesions, autoantibodies against Ro/La ribonucleoproteins, rheumatoid factor, C4-hypocomplementemia, cryoglobulinemia, hypergammaglobulinemia, leukopenia, lymphadenopathy, persistent salivary gland enlargement, purpura, arthralgias, Raynaud’s phenomenon, EULAR SS disease activity index (ESSDAI) > 5, high risk of developing NHL, and NHL itself.

## 3. Discussion

Our findings support that the Akt/mTOR pathway is implicated in the pathogenesis of pSS. The phosphorylated Akt at both T308 and S473 residues and its downstream targets, namely, the phosphorylated at S319 FoxO1 transcription factor and the phosphorylated at T246 PRAS40, were found to be strongly expressed in the epithelia and infiltrating mononuclear cells at the MSGs of pSS patients, but not in sicca-controls, suggesting that the Akt/mTOR pathway is specifically activated in pSS and represents a uniform feature of the disease. This agrees with previous studies implicating the activation of the Akt pathway in other autoimmune diseases. Indeed, the synovium of patients with rheumatoid arthritis and the affected skin of scleroderma patients exhibited increased activation of mTOR compared to osteoarthritis, or the normal skin, respectively, whereas the administration of rapamycin, an Akt/mTOR pathway inhibitor, in lupus patients and experimental models lowered the proinflammatory T-cell subsets by reverting the expansion of Th17 and upregulating Tregs [23,24,25,26,27,28].

The factors triggering Akt activation in SS are not known. Cytokines and growth factors are known to stimulate the Akt signaling cascade [11]. Thus, it would be tempting to hypothesize that Akt activation in pSS associates exclusively with the inflammatory milieu. In support of this, it appears that IFN signaling, known to be upregulated in MSGs of pSS patients [29,30], and Akt activation are reciprocally regulated. It has been shown that IFN signaling can induce inflammatory responses through Akt activation, whereas Akt1 mediates the full activation of IFN-regulated gene expression [31,32]. However, this is probably not the case, since the staining pattern was similar throughout the MSG tissue and did not associate with proximity to inflammatory lesions, whereas strong expression was also noticed in patients with restricted/mild infiltration. Furthermore, it does not associate with the predominant immune responses, since similar expression was noticed in MSGs of patients with mild or severe lesions, where T or B cells predominate, respectively [3]. Hence, although inflammation and the resultant local microenvironment may participate in the activation of Akt signaling at MSGs of pSS patients, they probably do not represent a triggering factor, but a positive feedback signal.

The activation of the Akt pathway in epithelial cells requires further attention. Akt signaling has been linked to epithelial to mesenchymal transition (EMT) related to migration, invasiveness, and metastasis of epithelial tumors [33]. EMT has not been described in pSS, whereas solid tumors are rare. In an attempt to evaluate a possible link between inflammation and fibrosis in the MSG lesions of pSS, it has been shown in in vitro assays that interleukin (IL)-17, IL-22, and subsequent TGFβ1 production may promote EMT in SGECs [34,35]. These data suggest that epithelial Akt activation in pSS is not associated with tumorigenesis. As mentioned before, the strong expression of phosphorylated Akt and signaling targets in epithelial structures distant from inflammatory lesions, as well as in patients with mild infiltrates, suggests that epithelial Akt activation does not associate with inflammation. The activation of the Akt pathway in MSG epithelia most likely represents another manifestation and/or sequel of the intrinsic epithelial activation that characterizes pSS [9]. Despite extensive studies, the offending factor(s) of epithelial activation in pSS is still unknown; however, viral infections and/or epigenetic changes have been incriminated [9]. Interestingly, *Helicobacter pylori* infection has been strongly associated with the activation of the Akt pathway in gastric epithelial cells and related oncogenesis, including adenocarcinoma and MALT lymphoma [36,37,38,39,40,41,42]. We also confirmed the operation of this pathway in gastric epithelia from patients with *Helicobacter pylori* infection and associated MALT lymphoma. In this context, a viral or other microbial infection may trigger the activation of the Akt pathway in pSS. In favor of this, it has been reported that the PI3K-Akt pathway is activated by EGF, whereas it has been also shown to mediate TLR3-induced apoptosis of cultured SGECs [19,20], which seems to have a significant role in disease development [9,43,44,45]. 

Prompted by the fact that pSS is at the crossroads of autoimmunity and malignancy, our initial hypothesis was that the Akt pathway may be implicated in NHL development in pSS. This was based on: (a) the major role of the Akt pathway in hematologic malignancies [17], (b) its recent implication in the development of gastric NHL related to *Helicobacter pylori* infection [42], which is also considered to arise due to chronic aberrant antigenic stimulation, as pSS-associated NHLs, and (c) a previous study describing PI3Kδ signaling in infiltrating lymphocytes at the MSG lesions of pSS patients [22]. Intriguingly, our data do not support a major role for Akt signaling in pSS-related lymphomagenesis. A similar expression pattern of the Akt/mTOR pathway was observed in all pSS patient subgroups, independently from the risk of developing NHL or having lymphoma. Furthermore, it was not found to associate with any of the disease features examined, suggesting that this pathway is universally activated in pSS and cannot discriminate patient subsets. However, its participation in NHL development cannot be excluded. PI3K/Akt/mTOR activation may be an event taking place in the initial stages of a multistep process that requires further molecular and signaling alterations to induce the transition from autoimmunity to lymphoma. The enrichment of the autoimmune milieu with an active PI3K/Akt/mTOR pathway may act synergistically with other pathways towards lymphoma development. The latter is evident in the co-activation of PI3K/Akt and MYC-driven lymphomagenesis in Burkitt lymphoma, a germinal center B-cell-derived tumor [46].

Despite the inability of the Akt pathway to discriminate distinct pSS patient subsets or patients at high risk for lymphomagenesis, it was specifically expressed in pSS patients, rendering it an attractive target for therapeutic intervention. A wide range of the PI3K/Akt/mTOR pathway inhibitors have been developed and tested mainly in a variety of malignancies [11]. This inhibitory effect has been associated with promotion of the p27 Kip-1-mediated G1 cell-cycle arrest and autophagic control [47]. However, since these agents have considerable side effects, their application in a benign disease such as pSS should be under significant consideration. In this context, a study testing seletalisib, a selective PI3Kδ inhibitor [48], had to stop before reaching statistical reliability; however, a non-significant reduction of disease activity in pSS patients compared to placebo was observed [48]. 

To conclude, even though our study is purely descriptive, it strongly supports the specific activation of the PI3K/Akt/mTOR pathway in pSS and its implication in disease pathogenesis, rendering it an attractive target for therapeutic intervention. The signals and/or processes triggering the activation of this pathway in pSS remain to be defined.

## 4. Materials and Methods

### 4.1. Patients

Twenty-nine patients with pSS [49], nine of whom had NHL (SSL), ten sicca-complaining individuals with no infiltrates in the diagnostic MSG biopsy and negative autoantibody profile (sicca-controls; six women and one man with median age 45, range 43–70), and five patients with gastric mucosa-associated lymphoid tissue lymphomas (MALT) associated with Helicobacter pylori infection (positive control group) were studied. The pSS patients without evidence of NHL at the time of biopsy were stratified according to MSG lesion severity, as arbitrarily defined by focus (FS) and Tarpley (TS) biopsy scores [3], to mild (*n* = 5; median FS, range: 1.0, 1.0–1.67, TS: 1, focal infiltrates), intermediate (*n* = 5; FS: 2.6, 1.71–3.27, TS: 2, focal infiltrates), and severe lesions (*n* = 10; FS: 5.82, 3.6–10.44, TS: 3–4, diffuse infiltrates associated with severe acinar destruction and loss of tissue architecture). The pSS patients without evidence of NHL at the time of biopsy were further classified according to the model provided by Fragkioudaki et al. [50] to low (*n* = 10; median follow-up time 3.7 years, range: 1.0–20.0 years) and high risk (*n* = 10; median follow-up time 2.5 years, range: 1.0–21.0 years) of developing NHL. Briefly, patients expressing two or less of the following risk factors, including salivary gland enlargement, lymphadenopathy, Raynaud’s phenomenon, anti-Ro/SSA or/and anti-La/SSB autoantibodies, rheumatoid factor positivity, monoclonal gammopathy, and C4 hypocomplementemia, were characterized as low risk, whereas those with three or more as high risk of developing lymphoma. Four of the high-risk pSS patients developed lymphoma (three MALT and one nodal marginal zone lymphoma; NMZL) during their follow-up (pre-lymphoma, median time to lymphoma diagnosis 7.46 years, range 2.0–15.5 years). The pSS-associated NHLs consisted of six MALT (located at MSGs in four and parotid glands in two), one follicular, and two diffuse large B-cell lymphomas (DLCBL). The MSG biopsy was performed at SSL diagnosis in 1 or 3.5 years (median; range 1–15 years) after the SSL diagnosis. Four patients had received anti-CD20 treatment 2.5 years (median; range 1–7 years) prior to biopsy performance. The two DLBCLs had received treatment with R-CHOP (Rituximab, Cyclophosphamide, Doxorubicin hydrochloride, Vincristine, Prednisolone) 7 years before biopsy performance. Two pSS patients without evidence of NHL were treated with steroids prior to biopsy sampling and three patients with steroids and hydroxychloroquine. 

The histologic features of the MSG biopsies, including the lesion severity as described before, focus and Tarpley scores, formation of ectopic germinal centers (GCs), and T- or B-cell predominance, were recorded for all patients included. The presence of ectopic lymphoid structures in MSG lesions was evaluated by both microscopic evaluation of hematoxylin and eosin-stained sections and immunohistochemical detection of CD3+ -T, CD20+ -B, and CD21+ -follicular dendritic cells in serial sections as described in Section 4.3. The medical records were retrospectively evaluated for various clinical, laboratory, and histological parameters of SS and lymphoma, including ESSDAI, arthralgias, arthritis, Raynaud’s phenomenon, salivary gland enlargement (SGE), palpable purpura, vasculitis, lung involvement, as attested by pulmonary-function tests and X-ray and/or computed-tomography scans, renal involvement (persistent proteinuria/glomerular hematuria and verification by renal biopsy), liver involvement (liver biopsy indicative of primary biliary cirrhosis), peripheral neuropathy as attested by nerve-conduction studies, anti-Ro/SSA and/or anti-La/SSB autoantibodies, rheumatoid factor, hypocomplementemia (C4 < 16 mg/dL and C3 < 75 mg/dL), hypergammaglobulinemia (IgG gamma-globulins > 2 g/L), anemia (hemoglobin < 12 g/dL), leukopenia (white blood cell count < 4000/mm^3^), lymphopenia (lymphocyte count < 1000/mm^3^), and neutropenia (neutrophil count < 1500/mm^3^). Patients’ characteristics are summarized in Table 1.

The study was approved by the Ethics Committee of School of Medicine, National and Kapodistrian University of Athens, Greece (Protocol No: 489, 11 March 2021), and the samples were used after informed consent, following the general data protection regulations (GDPR) of the European Union and the Helsinki Declaration principles.

### 4.2. Antibodies

Antibodies against human Akt (rabbit monoclonal; mAb, clone C67E7, working dilution: 1/300), FoxO1 (rabbit mAb, clone C29H4, working dilution: 1/200), and PRAS40 (rabbit mAb, clone D23C7, working dilution: 1/500) were from Cell Signaling Technology (Danvers, MA, USA). The antibody against human Akt phosphorylated at T308 (phosphoAkt-T308; mouse mAb, working dilution: 1/25) was purchased from OriGene (Rockville, MD, USA), whereas the one recognizing the phosphorylated at S473 (phosphoAkt-S473) molecule (rabbit mAb, clone EP2109Y, working dilution: 1/150) was from Abcam (Cambridge, UK). The antibodies to phosphorylated FoxO1 (phosphoFoxO1-S319, rabbit polyclonal, working dilution: 1/1000), PRAS40 (phosphoPRAS40-T246, rabbit polyclonal, working dilution: 1/200), and CD21 (rabbit mAb, EP3093, working dilution: 1/500) were also from Abcam. The antibody to CD3 (rabbit mAb, working dilution: 1/500) was from Cell-Marque (Rocklin, CA, USA), whereas antibodies to CD20 (mouse mAb L26, working dilution: 1/200) and cytokeratin (mouse mAb, clone MNF116, working dilution: 1/400) were from Dako (Glostrup, Denmark). The EnVision system (Dako) recognizing mouse and rabbit antibodies was employed as a second antibody and development system. 

### 4.3. Immunohistochemical Detection of the Molecules Implicated in the Akt Signaling Pathway

Following optimization, a standard immunohistochemical technique [3] was applied for the detection of all the aforementioned molecules. Briefly, MSG sections (4 μm) were deparaffinized, blocked for endogenous peroxidase activity by a 20 min incubation in 0.5% H2O2 and antigens were retrieved by microwaving in Tris/EDTA solution, pH: 9.0, for 15 min. Non-specific antibody binding was blocked by incubation with TBS buffer supplemented with 10% normal non-immune fetal bovine serum for 15 min and incubation with primary antibodies was performed overnight at 4 °C. Negative controls included staining with irrelevant isotype-matched antibodies and no addition of primary antibody. Stomach biopsies from five patients with gastric MALT lymphomas associated with Helicobacter pylori infection served as positive control tissues. The standardization of the staining procedure, including antigen retrieval technique, blockade of non-specific binding, and antibody dilutions were performed in a lymph node biopsy from a patient with DLBCL and tonsil. To evaluate the types of cells expressing the Akt forms and downstream molecules, staining with anti-CD3, anti-CD20, and anti-cytokeratin antibodies was routinely performed in serial sections and served for the identification of T cells, B cells, and epithelial cells, respectively. Furthermore, these stainings served as a positive control evaluating the quality of immunohistochemistry in each MSG tissue (CD3 and CD20 in pSS and cytokeratin in all MSG tissues).

The signal intensity of phosphoAkt-S473, representing the fully activated Akt, was analyzed using ImageJ Fiji software (National Institutes of Health, Bethesda, MD, USA) in serial images covering the entire area of MSG biopsies that were acquired by the computerized image-acquisition Axioskop ProgRes CapturePro software (Jenoptik-Laser, Optik-Systeme, Jena, Germany) connected to an Axioskop-40 microscope (Carl Zeiss, Thornwood, NY, USA) at ×10 objective magnification. The staining intensity of phosphoAkt-S473 normalized by the nuclei intensity value was quantified according to the protocol provided by Crowe and Yue [51].

### 4.4. Statistical Analyses

Differences among pSS patients with or without NHL and sicca-controls were evaluated by the non-parametric Kruskal–Wallis test, whereas differences between patients expressing or not various clinical, histological, and serological markers were analyzed by the non-parametric Mann–Whitney test. Associations with continuous values were evaluated by Spearman’s rank correlation test. Statistical significance was defined as a *p*-value of less than 0.05 for all comparisons; *p*-values were 2-tailed. Only the statistically significant differences are reported. GraphPad Prism-6 (GraphPad Software, San Diego, CA, USA) software was used.

## Figures and Tables

**Figure 1 ijms-22-13441-f001:**
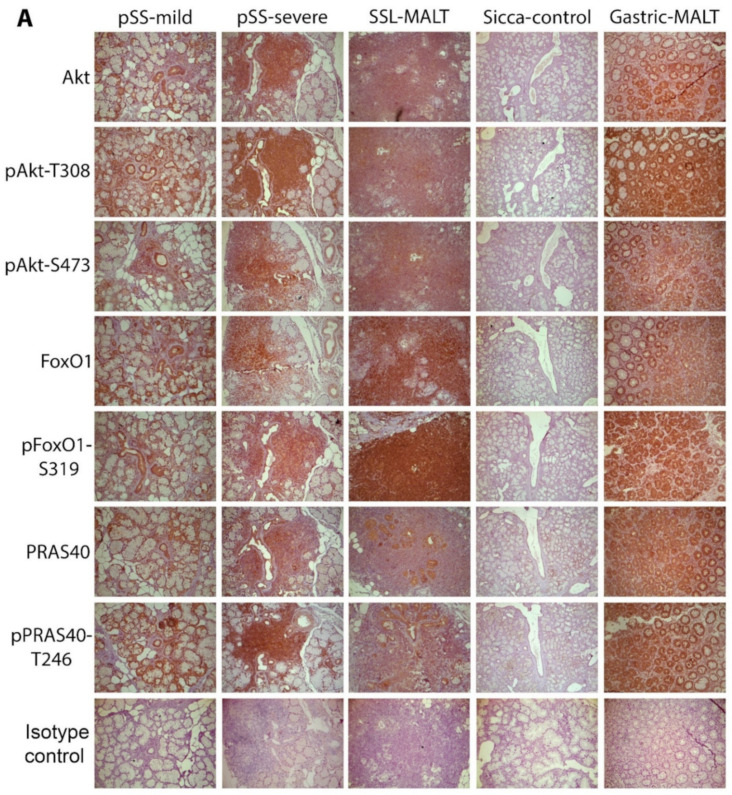
Akt signaling pathway is activated in the MSGs of pSS patients, but not in sicca-complaining controls. (**A**) Representative pictures of the immunohistochemical detection of Akt, phosphoAkt-T308 *(pAkt-T308)*, phosphoAkt-S473 *(pAkt-S473)*, FoxO1, phosphoFoxO1-S319 *(pFoxO1-S319)*, PRAS40, and phosphoPRAS40-T246 *(pPRAS40-T246)* molecules and isotype control in the MSGs of pSS patients and sicca-complaining controls *(sicca-control)*, as well as in the stomach of patients with MALT lymphoma associated with H. pylori infection *(gastric-MALT)* showing strong expression in patients with pSS or gastric MALT, but not in sicca-controls. (**B**) Detail of the pictures of (**A**) representing staining of the indicated molecules in pSS patients and sicca-controls. Similar expression of all the tested molecules was observed among pSS subgroups, including pSS patients without NHL, with mild or severe inflammatory infiltrates in the MSGs (*pSS-mild and pSS-severe,* respectively) or with NHL (*SSL*; the figures shown are from an SSL patient with MALT lymphoma, *SSL-MALT*). Original objective magnification: ×20.

**Figure 2 ijms-22-13441-f002:**
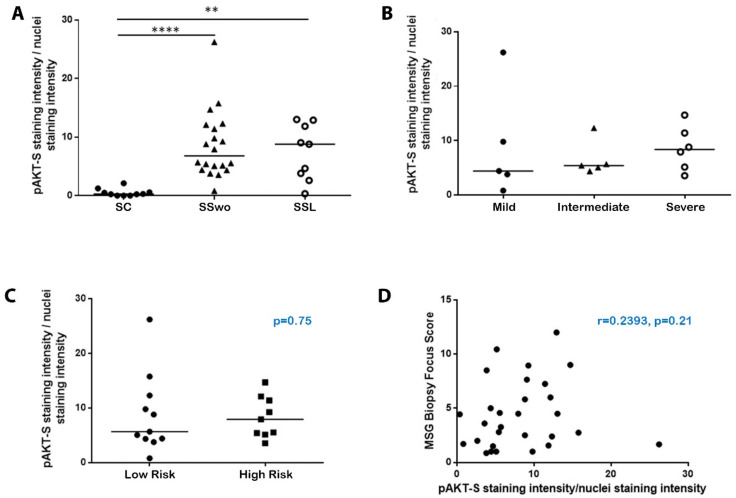
Akt is fully activated in the MSGs of pSS patients but does not associate with any histological or other disease features. (**A**) Dot plot displaying Kruskal–Wallis analysis of the staining intensity of the fully activated S473-phosphorylated Akt (pAKT-S) normalized by the nuclei staining intensity in MSGs from sicca-complaining controls (SC; black dots), pSS patients without evidence of NHL (SSwo; black triangles), and pSS patients with NHL (SSL; hollow dots). *p*-values are designated by asterisks (**: *p* < 0.01, ****: *p* < 0.0001), whereas horizontal bars represent the mean value of the group. Only statistically significant associations are indicated. (**B**) Kruskal–Wallis analysis revealed similar staining intensity of the S473-phosphorylated Akt (pAKT-S) normalized by the nuclei staining intensity in the MSGs of pSS patients with mild, intermediate, or severe lesions. Horizontal bars represent the mean value of the group. (**C**) Mann–Whitney non-parametric analysis displayed that the staining intensity of the S473-phosphorylated Akt (pAKT-S) normalized by the nuclei staining intensity in the MSGs did not differ between pSS patients at low (dots) or high risk (squares) to develop lymphoma (median intensity score 5.67 and 7.93, respectively; *p* = 0.75). (**D**) Spearman’s rank correlation analysis did not reveal any association between the staining intensity of the S473-phosphorylated Akt (pAKT-S) normalized by the nuclei staining intensity in the MSGs and biopsy focus score.

**Table 1 ijms-22-13441-t001:** Characteristics of the pSS patients studied. These included patients (a) at low risk of developing NHL (pSS low-risk), (b) at high risk (pSS high-risk), and (c) having NHL (SSL).

Features		pSS Patients		
		pSS Low-Risk (*n* = 10)	pSS High-Risk (*n* = 10)	SSL (*n* = 9)
**General**	Age (years), median (range)	60 (38–72)	53.5 (27–71)	71.5 (58–77)
	Men/women	1/10	0/10	1/9
	Disease duration (years), median (range)	11.5 (2.0–21.0)	4.5 (2.0–16.0)	13.0 (3.0–37.0)
**Histological** *(MSG biopsy)*	Biopsy focus score *(number of lymphocytic foci/4 mm^2^**)*, median (range)	1.67 (1.0–3.27)	4.57 (2.75–10.44)	4.44 (1.5–12.0)
	Germinal center formation, No (%)	2(20.0)	4(40.0)	3(33.3)
**Clinical**	Arthralgias, No (%)	6(60.0)	5(50.0)	7(77.8)
	Arthritis, No, (%)	2(20.0)	3(30.0)	2(22.2)
	SG enlargement (SGE), No (%)	0(0.0)	6(60.0)	7(77.8)
	Raynaud’s phenomenon, No (%)	3(30.0)	2(20.0)	2(22.2)
	Lymphadenopathy, No (%)	1(10.0)	2(20.0)	2(22.2)
	Parenchymal organ involvement, No (%)	2(20.0)	2(20.0)	1(11.1)
	*Lung involvement, No (%)*	1(10.0)	2(20.0)	1(11.1)
	*Renal involvement, No (%)*	0(0.0)	0(0.0)	0(0.0)
	*Liver involvement, No (%)*	1(10.0)	0(0.0)	0(0.0)
	Indicative of vasculitic involvement, No (%)	0(0.0)	2(20.0)	4(44.4)
	*Palpable purpura, No (%)*	0(0.0)	1(10.0)	2(20.0)
	*Vasculitis, No (%)*	0(0.0)	2(20.0)	0(0.0)
	*Glomerulonephritis, No (%)*	0(0.0)	0(0.0)	0(0.0)
	*Peripheral neuropathy, No (%)*	0(0.0)	2(20.0)	2(22.2)
	ESSDAI score, median (range)	3(0–5)	10.5(5–20)	17(15–22)
**Laboratory**	Anti-Ro/SSA and/or La/SSB positive, No (%)	9(90.0)	9(90.0)	7(77.8)
	*Anti-Ro/SSA positive, No (%)*	*9(90.0)*	*9(90.0)*	*7(77.8)*
	*Anti-La/SSB positive, No (%)*	*3(30.0)*	*8(80.0)*	*4(44.4)*
	Rheumatoid Factor positive, No (%)	4(40.0)	8(80.0)	8(88.9)
	C4-hypocomplementemia, No (%)	0(0.0)	7(70.0)	7(77.8)
	Cryoglobulinemia, No (%)	0(0.0)	3(30.0)	6(66.7)
	Leukopenia, No, (%)	1(10.0)	1(10.0)	1(11.1)

## Data Availability

Data sharing not applicable.

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
