# Peer review of "Akt Signaling Pathway Is Activated in the Minor Salivary Glands of Patients with Primary Sjögren’s Syndrome"

_ijms, 2021, doi:10.3390/ijms222413441_

Round 1

Reviewer 1 Report

The article is well written. There are not word mistakes. Overall, they provide a good job.
I suggested minor points which can improve this article to be better one.

[Introduction] 
I think “sicca” should be written in “SICCA” because this word is proper noun. Moreover, you should explain what is “SICCA” in introduction section. You also have to write the proper words (non-abbreviated) in the first appearance of the article.  

L69-71; You can also cite the article below, because this paper can explain the efficacy of mTORC2 inhibition for the late-stage oral squamous cell carcinoma treatment.

Naruse T, Yanamoto S, Okuyama K, Yamashita K, Omori K, Nakao Y, Yamada SI, Umeda M. Therapeutic implication of mTORC2 in oral squamous cell carcinoma. Oral Oncology 2017;65:23-32.

[Results]
L107; As the “MNC” is the first appearance word, you have to represent what it is.

Figure2B; I think the contents of horizontal line are hard to understand. You should represent only “mild”, “intermediate”, and “severe” in the graph. You already explain the degree of the SS lesion in L150-151. In Figure2C, you represent only “Low Risk” and “High Risk”, are easy to understand.  
Figure2C; You should add p-value of the Mann-Whitney U test.

L158; What is “ESSD AI”? Please also add non-abbreviated style.

[Discussion]
L232-234; It is reported that the inhibition of Akt/mTOR pathway induces cell cycle arrest at G1 phase and then inhibits cell migration. And this is also related in autophagic control in cells. This phenomenon is important sight when you discuss about this pathway. It is better of your article to cite the article below. 

Okuyama K, Suzuki K, Naruse T, Tsuchihashi H, Yanamoto S, Kaida A, Miura M, Umeda M, Yamashita S. Prolonged cetuximab treatment promotes p27 Kip1-mediated G1 arrest and autophagy in head and neck squamous cell carcinoma. Scientific Reports 2021;11(1):5259.

[Materials and Methods]
The concentrations of antibodies used should be represented.

Author Response

The article is well written. There are not word mistakes. Overall, they provide a good job.
I suggested minor points which can improve this article to be better one.

We sincerely thank you for your nice comments, the meticulous evaluation and your effort towards the improvement of our manuscript. Your contribution to the improvement of our manuscript is highly appreciated.

[Introduction] 
I think “sicca” should be written in “SICCA” because this word is proper noun. Moreover, you should explain what is “SICCA” in introduction section. You also have to write the proper words (non-abbreviated) in the first appearance of the article.  

“Sicca” is a latin word meaning dryness and is widely used for the description of mouth and eye dryness in primary Sjogren’s syndrome (sicca symptoms). This definition was added in introduction (L36-L38). The term “SICCA” in capital represents a patient cohort (also reported in the 2016 criteria).

We tried our best to address the missing definitions of abbreviations at their first appearance in the text. We are sorry for this omission.

L69-71; You can also cite the article below, because this paper can explain the efficacy of mTORC2 inhibition for the late-stage oral squamous cell carcinoma treatment.

Naruse T, Yanamoto S, Okuyama K, Yamashita K, Omori K, Nakao Y, Yamada SI, Umeda M. Therapeutic implication of mTORC2 in oral squamous cell carcinoma. Oral Oncology 2017;65:23-32.

Thank you very much for this suggestion. The reference was added in L73, Ref. No 18.

[Results]
L107; As the “MNC” is the first appearance word, you have to represent what it is.

The definition of the “MNC” abbreviation is reported on L96.

Figure2B; I think the contents of horizontal line are hard to understand. You should represent only “mild”, “intermediate”, and “severe” in the graph. You already explain the degree of the SS lesion in L150-151. In Figure2C, you represent only “Low Risk” and “High Risk”, are easy to understand.  
Figure2C; You should add p-value of the Mann-Whitney U test.

All suggestions were incorporated in Figure 2.

L158; What is “ESSDAI”? Please also add non-abbreviated style.

We are sorry for this omission. ESSDAI represents the EULAR SS disease activity index. This information was added at the text (L157).

[Discussion]
L232-234; It is reported that the inhibition of Akt/mTOR pathway induces cell cycle arrest at G1 phase and then inhibits cell migration. And this is also related in autophagic control in cells. This phenomenon is important sight when you discuss about this pathway. It is better of your article to cite the article below. 

Okuyama K, Suzuki K, Naruse T, Tsuchihashi H, Yanamoto S, Kaida A, Miura M, Umeda M, Yamashita S. Prolonged cetuximab treatment promotes p27 Kip1-mediated G1 arrest and autophagy in head and neck squamous cell carcinoma. Scientific Reports 2021;11(1):5259.

Thank you for this suggestion. Both the comment and the reference were incorporated in the discussion section (L234-235 and Reference No 48, respectively).

[Materials and Methods]
The concentrations of antibodies used should be represented.

In some of the antibodies, the concentration or the amount of the antibody was not provided by the manufacturer. Therefore, we added the working dilution of each antibody in “Antibodies” section of Materials and Methods (first paragraph of page 10).

Reviewer 2 Report

The manuscript by Stergiou et al. entitled “Akt signaling pathway is activated in the minor salivary glands 2 of patients with primary Sjögren’s syndrome” demonstrated that higher activation of Akt was found on lymphocyte and epithelial cells of minor salivary gland than that in non-SS patients.

The phosphorylated Akt and its substrates, PRAS40 and FoxO1, were strongly expressed in the minor salivary gland epithelial and infiltrating mononuclear cells of pSS patients, but sicca-control. It seems that the present results are consistent with the previous reports by Nakamura et al. (Rheumatol Int, 2007).

Major points:

  1. In Figure 1, immunohistochemical findings were shown. It is difficult to understand which cells are positive for each factor. HE stainings and enlarged images should be included. In addition, the negative control should be also added.
  2. The signal intensity of phosphoAkt-S473 was described to be analyzed using ImageJ. Was the intensity of epithelial cells and lymphocytes analyzed individually?

Author Response

The manuscript by Stergiou et al. entitled “Akt signaling pathway is activated in the minor salivary glands 2 of patients with primary Sjögren’s syndrome” demonstrated that higher activation of Akt was found on lymphocyte and epithelial cells of minor salivary gland than that in non-SS patients.

The phosphorylated Akt and its substrates, PRAS40 and FoxO1, were strongly expressed in the minor salivary gland epithelial and infiltrating mononuclear cells of pSS patients, but sicca-control. It seems that the present results are consistent with the previous reports by Nakamura et al. (Rheumatol Int, 2007).

Thank you very much for your comments and the careful evaluation of our manuscript. Indeed, our results are consistent with the publication by Nakamura et al showing that epidermal growth factor activates PI3K-Akt pathway in salivary epithelial cells. This information was incorporated in the introduction and discussion sections (L74-76 and L207-208, Reference No 19)

Major points:

  1. In Figure 1, immunohistochemical findings were shown. It is difficult to understand which cells are positive for each factor. HE stainings and enlarged images should be included. In addition, the negative control should be also added.

The negative control stainings were added in Figure 1A. Furthermore, a detail of each panel of Figure 1 (regarding pSS patients and sicce-controls) was added in Figure 1B. The HE stainings are presented in supplementary Figure 1.

  1. The signal intensity of phosphoAkt-S473 was described to be analyzed using ImageJ. Was the intensity of epithelial cells and lymphocytes analyzed individually?

The staining intensity of phosphoAkt-S473 was analyzed by ImageJ in the entire tissue. In preliminary analyses (2 samples from each patient group), we analyzed the staining intensity score of epithelial cells and lymphocytes separately. Similar staining intensity scores were obtained, confirming the microscopic evaluation of all samples by EKK and MV. Since the separate analysis of epithelial and infiltrating mononuclear cells was time-consuming and did not reveal any differences, we performed the analysis of staining intensity without discrimination in the entire tissue (at least four glands) of all samples.

Round 2

Reviewer 2 Report

The authors have done what I have requested.